# Genomic Confirmation of the P-IIIe Subclass of Snake Venom Metalloproteinases and Characterisation of Its First Member, a Disintegrin-Like/Cysteine-Rich Protein

**DOI:** 10.3390/toxins14040232

**Published:** 2022-03-23

**Authors:** Kity Požek, Adrijana Leonardi, Jože Pungerčar, Weiqiao Rao, Zijian Gao, Siqi Liu, Andreas Hougaard Laustsen, Alenka Trampuš Bakija, Katarina Reberšek, Helena Podgornik, Igor Križaj

**Affiliations:** 1Department of Molecular and Biomedical Sciences, Jožef Stefan Institute, SI-1000 Ljubljana, Slovenia; kity.pozek@ijs.si (K.P.); adrijana.leonardi@ijs.si (A.L.); joze.pungercar@ijs.si (J.P.); 2Department of Biotechnology and Biomedicine, Technical University of Denmark, DK-2800 Kongens Lyngby, Denmark; weirao@dtu.dk (W.R.); ahola@bio.dtu.dk (A.H.L.); 3Wuxi Fisheries College, Nanjing Agricultural University, Wuxi 214128, China; yuan08102021@126.com; 4Department of Mass Spectrometry, Beijing Genomics Institute-Research, Shenzhen 508083, China; siqiliu@genomics.cn; 5Division of Pediatrics, University Medical Centre Ljubljana, SI-1000 Ljubljana, Slovenia; alenka.trampus@kclj.si; 6Department of Hematology, University Medical Centre Ljubljana, SI-1000 Ljubljana, Slovenia; kc34446@ukclj.si (K.R.); helena.podgornik@kclj.si (H.P.); 7Department of Clinical Biochemistry, Faculty of Pharmacy, University of Ljubljana, SI-1000 Ljubljana, Slovenia

**Keywords:** snake venom, *Vipera ammodytes*, disintegrin-like/cysteine-rich protein, gene structure, platelet aggregation, snake venom metalloproteinase (SVMP)

## Abstract

Disintegrin-like/cysteine-rich (DC) proteins have long been regarded just as products of proteolysis of P-III snake venom metalloproteinases (SVMPs). However, here we demonstrate that a DC protein from the venom of *Vipera ammodytes* (*Vaa*; nose-horned viper), VaaMPIII-3, is encoded per se by a P-III SVMP-like gene that has a deletion in the region of the catalytic metalloproteinase domain and in part of the non-catalytic disintegrin-like domain. In this way, we justify the proposal of the introduction of a new subclass P-IIIe of SVMP-derived DC proteins. We purified VaaMPIII-3 from the venom of *Vaa* in a series of chromatographic steps. A covalent chromatography step based on thiol-disulphide exchange revealed that VaaMPIII-3 contains an unpaired Cys residue. This was demonstrated to be Cys6 in about 90% and Cys19 in about 10% of the VaaMPIII-3 molecules. We further constructed a three-dimensional homology model of VaaMPIII-3. From this model, it is evident that both Cys6 and Cys19 can pair with Cys26, which suggests that the intramolecular thiol-disulphide exchange has a regulatory function. VaaMPIII-3 is an acidic 21-kDa monomeric glycoprotein that exists in at least six *N*-glycoforms, with isoelectric points ranging from pH 4.5 to 5.1. Consistent with the presence of an integrin-binding motif in its sequence, SECD, VaaMPIII-3 inhibited collagen-induced platelet aggregation. It also inhibited ADP- and arachidonic-acid-induced platelet aggregation, but not ristocetin-induced platelet agglutination and the blood coagulation cascade.

## 1. Introduction

Snake venoms are complex mixtures of toxic and non-toxic substances that snakes use to catch and immobilise their prey or as weapons for self-defence [1]. The most venomous European snake is the nose-horned viper, *Vipera ammodytes (Vaa)*, the venom of which causes haemotoxic, neurotoxic, myotoxic, and cardiotoxic effects in mammals [2,3,4,5]. Some of the most abundant toxins in viperid venoms are snake venom metalloproteinases (SVMPs). These have evolved through domain loss and have diversified in parallel, structurally and functionally, through post-translational processing [6].

SVMPs are grouped into three main classes (P-I, P-II, P-III) and several subclasses based on the structure of the nascent gene products and their post-translational processing and modifications [7]. The P-III SVMPs are synthesised as precursors that consist of a prodomain, a catalytic metalloproteinase (MP) domain, and two non-catalytic domains: a disintegrin-like (D) and a cysteine-rich (C) domain. The P-III class is evolutionarily the oldest class of the SVMPs. It is likely to have evolved from an ancient disintegrin and metalloproteinase (*ADAM*) gene recruited into the venom gland of the advanced snakes (Caenophidia) [8,9]. The other two SVMP classes arose from the primordial P-III class gene, first by the loss of the C domain (P-II), and subsequently also by the loss of the D domain (P-I). In contrast, a recently discovered SVMP-like protein in *Vaa* venom, named VaaMPIII-3, has been proposed to result from the loss of the MP domain and the N-terminal part of the D domain, thus representing the new P-IIIe subclass of SVMPs [10]. After their post-translational processing, the precursors (pro-proteins) of SVMPs belonging to this subclass lead to the mature proteins that consist of a partial (D’) or complete D domain and a C domain, and are termed D’C or DC proteins.

In addition to VaaMPIII-3, several other DC proteins have been isolated from viperid snake venoms [11,12,13,14,15,16,17,18]. These have shown various pathophysiological effects ex vivo, such as inhibition of platelet aggregation induced by ADP, collagen, and arachidonic acid, inhibition of platelet agglutination induced by ristocetin, chemotactic recruitment of neutrophils, and stimulation of leukocyte rolling in the microcirculation. It is widely accepted that these are products of autoproteolytic processing of P-III SVMPs into their MP and DC parts [6]. This has relied on reports that some P-III SVMPs can be autoproteolytically processed in vitro to DC proteins [19,20,21,22,23,24,25,26,27]. Only one autoproteolytic product, acucetin, has been tested for its effects on platelets, where it was shown to inhibit platelet aggregation induced by ADP and collagen. The acknowledged structural determinant of such DC protein activity is the integrin-binding sequence SECD, which is located in the proteins’ D domain [28]. It is likely that the so-called ‘hypervariable region’ (HVR) in the proteins’ C domain is also important for their biological function [29,30,31].

The origin of the DC proteins in snake venoms is still controversial. It was shown that it is unlikely that autolysis of P-III SVMPs to DC and MP fragments occurs in the venom gland [7]. DC proteins might instead be products of proteolysis of parental P-III SVMPs by other proteases in the venom. In some venoms, however, only the DC fragment is found, and not also the corresponding MP fragment [7], which would imply total degradation of the MP fragment. From an evolutionary perspective, this would appear to be an irrational and unlikely solution. The possible third way of naissance of the DC proteins emerged from the analysis of the coding sequences of DC proteins in a cDNA library of the *Vaa* venom gland (VaaMPIII-3; cDNA, NCBI GenBank accession number MG958499) [10] and of the venom gland of *Echis carinatus sochureki* (cDNA, NCBI GenBank accession number GU012129) [6]. These sequences were suggested to be the transcription products of the DC protein genes.

Here, we report experimental confirmation of the ‘DC gene’ hypothesis. By describing the gene structure of the precursor for the *Vaa* venom DC protein VaaMPIII-3, we justify our proposal of the introduction of the P-IIIe subclass into the SVMP classification [10]. We also describe the basic physicochemical and biochemical properties and the biological functions of VaaMPIII-3, the first member of this subclass of SVMPs.

## 2. Results

### 2.1. VaaMPIII-2 and VaaMPIII-3 Gene Structures

VaaMPIII-2 was chosen for comparison of its gene structure with that of VaaMPIII-3 because it belongs to the ‘typical length’ P-III SVMPs, which have all three of the characteristic domains, namely, MP, D, and C. Moreover, in addition to VaaMPIII-3, we also detected VaaMPIII-2 at the protein level in a recent proteomic analysis [10]. It appears to belong to the P-IIId subclass of SVMPs, with a covalently bound two-chain snaclec subunit at its C-terminal end [3].

The full-length pre-pro-protein coding sequences of VaaMPIII-2 (cDNA, NCBI GenBank accession number MG958498) and VaaMPIII-3 (cDNA, NCBI GenBank accession number MG958499) were used to screen the first draft of the *Vaa* genome (W. Rao, Z. Gao, J. Pungerčar, S. Liu, and A.H. Laustsen, unpublished). An absolute match (i.e., 100% nucleotide identity in the exon regions) that started with the initiation codon (ATG) and ended with a stop codon (TAA) was found in contig 132, in the nucleotide sequence regions, from 2,699,183 to 2,729,959 base pairs for VaaMPIII-2, and from 2,308,964 to 2,332,096 base pairs for VaaMPIII-3. Both of these genes are in the same orientation, with the VaaMPIII-3 gene much shorter, by 7644 base pairs. Although the VaaMPIII-2 gene consists of 17 exons and 16 introns, and the VaaMPIII-3 gene consists of ten exons and nine introns, the length of the matching exons and the positions of the introns are very similar (Appendix A). More precisely, introns 1 to 6 and 7 to 9 in the VaaMPIII-3 gene correspond to introns 1 to 6 and 14 to 16, respectively, in the longer VaaMPIII-2 gene. In both genes, these comparable introns interrupt the protein coding sequence at identical positions, and are also in the same intron phase 0, 1, or 2. The nucleotide and amino acid sequences of VaaMPIII-2 and VaaMPIII-3 are shown in Figure 1, together with the positions of the mapped introns. The main difference is a large deletion of seven exons and seven introns in the middle part of the VaaMPIII-3 gene (Figure 2), which corresponds to the region from exon 7 to 13 in the VaaMPIII-2 gene, where a nucleotide sequence that encodes the catalytic MP domain of a ‘typical length’ P-III SVMP would be expected.

### 2.2. Isolation of VaaMPIII-3 from Crude Venom

VaaMPIII-3 was isolated from crude *Vaa* venom in three chromatographic steps (Figure 3). After initial fractionation of the venom by gel filtration, the B2 fraction containing proteins of 20–30 kDa was further separated by cation-exchange chromatography at pH 6. VaaMPIII-3 was eluted in the unbound fraction, which indicated an isoelectric point (pI) < 6. As a final purification step, covalent binding to Thiol-Sepharose 4B via the thiol group of the unpaired Cys residue in VaaMPIII-3 was exploited. To check the homogeneity of the isolated VaaMPIII-3, proteins retained by Thiol-Sepharose 4B were first filtered through a 10-kDa-cut-off filter in the presence of a reducing agent, dialysed and concentrated, and then subjected to Edman sequencing. Part of the sample was also fragmented with trypsin and analysed by liquid chromatography-electrospray ionisation-tandem mass spectrometry (Appendix A). Both of these methods confirmed the presence of only VaaMPIII-3 in the sample. Although the major N-terminal sequence was RAGTE (90%), another minor N-terminal sequence was detected (AGTE; 10%), as the main N-terminal sequence with Arg removed. As calculated, VaaMPIII-3 represented less than 1% (m/m) of all of the *Vaa* venom proteins.

### 2.3. Physicochemical Characterisation of VaaMPIII-3

On non-reducing SDS-PAGE gels, VaaMPIII-3 migrated predominantly as a monomer, with an apparent molecular mass of 21 kDa. A faint band that corresponded to a VaaMPIII-3 dimer was apparent, with a molecular mass of ~35 kDa (Figure 4a). The experimental mass of the monomer (mass spectrometry analysis is shown in Appendix A) is higher than the theoretical mass calculated from the amino acid sequence of VaaMPIII-3. The mass difference was shown to be due to the presence of protein-linked carbohydrates, as *N*-deglycosylation of VaaMPIII-3 using peptide *N*-glycosidase F showed that the molecule contains ~4 kDa of *N*-linked glycans (Figure 4b). Examination of the amino acid sequence of VaaMPIII-3 revealed only one potential *N*-glycosylation site, at position 64 (^64^NAT^66^), which thus represents the site where the carbohydrates are most likely to be attached to the protein.

As is characteristic for glycosylated proteins, the VaaMPIII-3 sample showed several pIs. These are from pH 4.5 to 5.1 (Figure 4c). Two-dimensional gel electrophoresis showed that there were six glycoforms of VaaMPIII-3 in the sample, of which the most acidic was also the most abundant (Figure 4d). We separated the most acidic isoform from the other isoforms by chromatography on a C-18 reversed-phase high-performance liquid chromatography (RP-HPLC) column (Figure 4e). The most acidic isoform of VaaMPIII-3 eluted in peak 1 (Figure 4f).

The structural thermostability of VaaMPIII-3 was investigated under different conditions by differential scanning fluorimetry (DSF). This showed that VaaMPIII-3 is relatively unstable in pure water (temperature of unfolding or melting (Tm), 47 °C), but more stable in various buffers at pH 5 to 9 (Tm, ≥60 °C) (Figure 5). The highest stability of VaaMPIII-3 was in 20 mM HEPES buffer at pH 7. This was additionally increased by addition of 2 mM Ca^2+^ ions (Tm, 72 °C). As expected, EDTA, a chelator of Ca^2+^ ions, negatively affected the stability. All of the other additives tested also destabilised VaaMPIII-3. Based on this study, we included 2 mM Ca^2+^ in buffers throughout the isolation procedure and during the activity testing of VaaMPIII-3 and avoided longer exposures of VaaMPIII-3 samples to destabilising chemicals, such as reducing agents (dithiothreitol, DTT) and high concentrations of NaCl.

### 2.4. Modelling of the Three-Dimensional Structure of VaaMPIII-3

We created a three-dimensional (3D) model of VaaMPIII-3 to define the spatial location of its unpaired Cys residue. The 3D model also revealed the position in space of the established structural determinant of function in DC proteins, the integrin-binding motif, and the HVR (Figure 6a). The crystal structure of a P-III SVMP from the venom of *Agkistrodon acutus* AaHIV (PDB code: 3HDB) was used as a template for homology modelling of VaaMPIII-3. The amino acid sequences of the overlapping parts of AaHIV and VaaMPIII-3 are 67% identical and 83% similar. The modelled 3D structure of VaaMPIII-3 includes structural Ca^2+^ ions as well as all of the hydrogen atoms. Justifying the choice of the crystal structure of AaHIV as a template for homology modelling of the 3D structure of VaaMPIII-3, the calculated root-mean-square deviation across all of the backbone residues was low, at 1.247 Å.

To additionally validate the 3D model of VaaMPIII-3, the data obtained were used to predict the circular dichroism (CD) spectrum of VaaMPIII-3, using the web server PDBMD2CD [32]. We then compared the calculated CD spectrum with the experimentally obtained far UV CD spectrum of VaaMPIII-3. The contents of the secondary structure elements in VaaMPIII-3 were calculated using the BeStSel tool [33]. The experimentally determined contents of the different types of secondary structure elements in VaaMPIII-3 were very similar to those calculated from the 3D homology model of VaaMPIII-3 (Table 1).

### 2.5. Determination of the Position of the Unpaired Cys Residue in VaaMPIII-3

The 3D model of VaaMPIII-3 based on the crystal structure of AaHIV indicated the unpaired Cys at position 19. To verify this, we carboxymethylated (CM) VaaMPIII-3 under mild reducing conditions (1.5 mM DTT), to avoid cleavage of the intrachain disulphide bonds while allowing for alkylation of the free Cys residues. By N-terminal Edman degradation of CM-VaaMPIII-3, we identified CM-Cys at positions 6 and 19, but not at position 13, which confirmed that the intrachain disulphide bonds remained intact under the experimental conditions used. Nine out of 10 VaaMPIII-3 molecules contained an unpaired Cys6 residue. The remaining VaaMPIII-3 molecules contained an unpaired Cys19 residue. According to the 3D model of VaaMPIII-3, both the Cys6 residue and the Cys19 residue can form a disulphide bond with the Cys26 residue. The distance between the Cys6 residue and the Cys26 residue in the 3D model is 7.309 Å, and the distance between the Cys19 residue and the Cys26 residue is 3.032 Å (Figure 6b). However, considering molecular dynamics, these Cys residues might come spatially closer than 3 Å, which would allow for the formation of the disulphide bond between adjacent Cys residues [34]. In conclusion, these Cys residues potentially undergo disulphide bond switching.

### 2.6. Functional Characterisation of VaaMPIII-3

VaaMPIII-3 inhibited platelet aggregation induced by ADP, collagen, and arachidonic acid (Figure 7a–c). The IC_50_ of VaaMPIII-3 for inhibition of ADP-induced platelet aggregation was 292 ± 15.433 nM. However, 0.7 µM VaaMPIII-3 had no effects on ristocetin-induced von Willebrand factor (vWF)-dependent platelet agglutination (Figure 7d). At 1 µM, VaaMPIII-3 also had no effects on blood coagulation times. Using flow cytometry, we demonstrated that VaaMPIII-3 did not affect platelet receptors GPIIb, GPIIIa, GPVI, and GPIX in terms of the proportions of antigen expression and mean fluorescence intensity (MFI). It should be noted, however, that negative results do not necessarily mean that binding does not occur in vivo, as detection is dependent on the fluorochrome, antibody clone, and flow cytometer settings. For GPIb-α (CD42b), the MFI was decreased by approximately 30% by VaaMPIII-3.

## 3. Discussion

To date, a number of DC proteins have been discovered and isolated from snake venoms (Table 2). Some of these have been shown or predicted to arise from P-III SVMPs by proteolysis [7]. P-III SVMPs that autolyse in vitro to DC and MP fragments require alkaline conditions, an absence of Ca^2+^, and slightly elevated temperature [19,20,21,22,23,24,25,26,27]. However, such conditions do not exist in venom glands [7]. It is possible that the DC protein is a specific cleavage product of the precursor P-III SVMP from another venom protease that is active in the venom gland, but convincing confirmation of such a mechanism is still not available. On the contrary, for snake venom DC proteins, such as VaaMPIII-3 [10], leberagin-C [11], BaltDC [18], catrocollastatin-C [17], and jararhagin-C [19], the precursor P-III SVMP molecules or the corresponding MP fragments have not been isolated from the venom.

In this study, we demonstrated experimentally that DC proteins can emerge in the venom after being synthesised de novo from their own genes. The highly similar structures of VaaMPIII-2 (P-IIId) and VaaMPIII-3 (P-IIIe) SVMP genes confirm that the mature DC protein lacking the MP domain is the result of a large deletion of the corresponding coding sequence already at the gene level, and not due to alternative pre-mRNA splicing. This is also consistent with our previous results, where all four isolated VaaMPIII-3 cDNAs (i.e., mRNA transcripts) had an identical nucleotide sequence with an unexpected deletion of 852 nucleotides that encode the MP domain and the first part of the subsequent D domain in the middle part of the genetic sequence (here referred to as the truncated D’ domain) [10].

The VaaMPIII-2 gene consists of 17 exons and 16 introns, and is one of the typical 17-exon P-III class SVMP genes for which close evolutionary relationships were well illustrated in a recent genome study of several *Crotalus* species of the Viperinae subfamily of viperid snakes [35]. This previous analysis has also shown that the SVMP family has expanded from a single and well-conserved 25-exon *adam28* gene to as many as 31 tandem genes in *Crotalus atrox*, arrayed in a single cluster of 1.3 Mb of genomic sequence, through a number of single-gene and multigene duplications and step-wise intragenic deletion events. The genomic deletions thus resulted in the appearance of all of the main classes of SVMPs in the viperid snakes (i.e., P-III, P-II, P-I; with evolution proceeding in this direction). It can also be observed that the loss of protein domains over the course of evolution was accompanied by a reduction in the number of exons, from 25 exons in the *adam28* gene to 17 exons in the P-III class, to 14 exons in the P-II class, and to 12 exons in the SVMP genes of the P-I class. In the *Vaa* genome, we also identified a shorter, P-III class-derived 10-exon gene of VaaMPIII-3, which is about 400 kb away from the ‘typical-length’ 17-exon P-III class VaaMPIII-2 gene. A more detailed analysis of this genomic segment in *Vaa* is still to be performed, but it is certain that in this case, the genomic deletion encompassed the entire catalytic MP domain, which led to the final production and accumulation of this unique D’C-domain protein in the viper venom after cleavage of the pro-peptide region. It also remains to be clarified whether a similar genomic event occurred in the case of a longer predicted DC domain P-III class SVMP from *Echis carinatus sochureki* venom (cDNA, NCBI GenBank accession number GU012129) [6]. As these last two snake species belong to the subfamily Viperinae, such a deletion of the MP domain in the P-III class proteins might either be specific to this subfamily of the snake family Viperidae or have evolved independently in these two species.

Alignment of the first 47 amino acid residues of the pre-pro-peptide region of VaaMPIII-3 (consisting of a 20-amino acid signal peptide, followed by 27 amino acids of a pro-peptide) shows high amino acid sequence similarity (>90%) with that of the other VaaMPIIIs, as well as with the pre-pro-region of the precursors of two putative homodimeric disintegrins from *Vaa*, VaaDis-1, and VaaDis-2 [10], which indicates their common origin. Such a homodimeric disintegrin can also form a heterodimer with a disintegrin derived from a P-II SVMP precursor (e.g., VaaMPII-1, VaaMPII-2, VaaMPII-3), the former being referred to as its α-subunit and the latter as its β-subunit. More specifically, heterodimeric disintegrins consist of two different α-subunits or one α-subunit and one β-subunit, while homodimeric disintegrins consist of two identical α-subunits. All these dimeric disintegrins have 10 Cys residues involved in the formation of four intrachain and two interchain disulphide bonds, and they probably evolved in the Viperidae from the P-II class SVMPs [36]. However, at least in *Vaa*, the pre-pro-peptide region of VaaMPIII-3 shows a somewhat lower similarity with that of VaaMPIIs (about 80% over the first 47 amino acid residues) than with Vaa-Dis (91–93%), which suggests that the α-subunits in this case might have evolved in an independent event from the P-III class SVMPs, albeit after deletion of the MP-coding and C-coding domains from their genes. In this respect, the pre-pro-, catalytic (MP), and non-catalytic (D and C) domains of SVMPs and their similar proteins, such as homodimeric disintegrins, might also have undergone different evolutionary paths in different viperid snakes.

The above conclusions about the evolution of different P-III SVMP genes in viperid snakes are limited to the experimental data obtained from the analysis of the *Vaa* draft genome, but are well supported by the transcriptomics and proteomics of *Vaa* venom. Nevertheless, additional data from related snake species are needed to elucidate the entire evolutionary history of this group of proteins in the *Viperidae* family.

We developed an efficient three-step procedure to purify VaaMPIII-3 from *Vaa* venom (Figure 3), which allows for the separation of its multiple isoforms. These are very likely *N*-glycoforms, as we have shown that the isolated 21-kDa VaaMPIII-3 carries a 4-kDa sugar moiety. The *N*-linked glycans lower the pI of VaaMPIII-3 from the theoretical 5.1 to 4.5 for its most abundant isoform. The *N*-linked glycans presumably stabilise the structure of VaaMPIII-3 and influence the activity of VaaMPIII-3 by modulation of its interaction with biological targets [37].

VaaMPIII-3 has 17 Cys residues in its structure, at least one of which is unpaired, which enabled binding of the protein to Thiol-Sepharose 4B, a property that we took advantage of in the final step of the purification procedure. Although VaaMPIII-3 has an unpaired Cys residue, it appears predominantly as a monomer in the venom, as demonstrated by the SDS-PAGE and RP-HPLC analyses. It appears that the unpaired Cys residue in VaaMPIII-3 is not prone to forming an intermolecular bond. To gain insight into the structure of VaaMPIII-3, we created a 3D homology model. As there is currently no DC protein with a resolved crystal structure, we chose the crystal structure of AaHIV from *A. acutus* snake venom as the homology modelling template, which is a full-length P-III SVMP protein with the highest amino acid sequence similarity (83%) to the overlapping region of VaaMPIII-3 [38]. The constructed 3D model of VaaMPIII-3 suggests that its Cys residue at position 19 is unpaired, as the corresponding Cys residue in AaHIV (Cys280) forms a disulphide bond with Cys254 in the N-terminal part of the D domain, which is absent in VaaMPIII-3. The only DC protein for which a disulphide bond pattern has been partially determined is catrocollastatin-C from *C. atrox* venom [39]. Unlike VaaMPIII-3, its D domain is not truncated. Although the disulphide bond pattern of catrocollastatin-C is different from that of AaHIV, its Cys residue corresponding to Cys19 in VaaMPIII-3 is also either unpaired or connected with a Cys residue in the N-terminal part of the D domain, not present in VaaMPIII-3. Experiments showed, however, that in VaaMPIII-3, the predominately unpaired Cys residue is Cys6 and not Cys19. According to the 3D model of VaaMPIII-3, the Cys6 residue should be paired with the Cys26 residue. An interesting observation is that the molar ratio is the same between unpaired Cys6 and Cys19 and between the N-terminal isoforms RAGTE and AGTE (1:9), implying that the RAGTE isoform contains an unpaired Cys6 residue, while the AGTE isoform contains an unpaired Cys19 residue. Nevertheless, the 3D model of VaaMPIII-3 offers another interesting hypothesis: from the model, it appears that Cys residues 6, 19, and 26 can come close enough to each other sterically to allow for mutual disulphide bond formation. This opens up the possibility that intramolecular thiol-disulphide exchange or disulphide switching can occur. This mechanism is known to be involved in the regulation of activity in proteins such as V-type ATPases and many others [40,41]. It is therefore possible that the biological activity of VaaMPIII-3 is regulated by intramolecular thiol-disulphide exchange or disulphide switching.

SVMPs are the major haemorrhagic substances in snake venoms. Studies of the DC proteins have revealed that it is not only the enzymatic activity of the MP domain that causes this pathology, but that the non-catalytic D and C domains are also important participants (Table 2). These domains, for example, interrupt the process of blood coagulation by specifically binding to receptors on platelets or to ligands of these receptors, such as collagen and vWF [42,43].

**Table 2 toxins-14-00232-t002:** Overview of some disintegrin-like/cysteine-rich proteins (DC proteins) isolated from snake venoms, and their inhibition of platelet agglutination/aggregation.

DC Protein	Snake	Inhibition		Reference
		Induced by	IC_50_ (nM)	
Alternagin-C	*Bothrops alternatus*	Collagen		[44]
Balt-DC	*Bothrops alternatus*	Epinephrine, ristocetin		[18]
BmooPAi	*Bothrops moojeni*	Ristocetin		[14]
Catrocollastatin-C	*Crotalus atrox*	Collagen	66	[17]
Halysetin	*Agkistrodon halys*	Collagen	420	[13]
Jararhagin-C	*Bothrops atrox*	Collagen	200	[12,45]
	*Bothrops jararaca*	ADP		
Jaracetin	*Bothrops jararaca*	Collagen		[46]
Leberagin-C	*Macrovipera lebetina*	Arachidonic acid	50	[11]
	*transmediterranea*	Thrombin	40	
VaaMPIII-3	*Vipera ammodytes*	Collagen, ADP, arachidonic acid		This study

We have shown that VaaMPIII-3 inhibits platelet aggregation induced by various agonists in human platelet-rich plasma, which implies that it interferes with multiple platelet activation pathways. Its effect on ADP-induced platelet aggregation has an IC_50_ of 292 nM, which is comparable to that of A/DC (IC_50_, 320 nM), a recombinant DC part of atrolysin A from the venom of *C. atrox* [47]. The C domain of atrolysin A alone (A/C) did not show this effect, which suggests that only its D domain is involved [48]. Jararhagin-C (isolated from the venom of *Bothrops jararaca*) and acucetin (released by autolysis of AaHIV) also affect this process, but their IC_50_ values were not determined [12,27]. However, not all DC proteins have this kind of inhibitory action [11,13,14,17,18]. ADP is an important aggregation agonist that triggers platelet activation by binding to P2Y receptors, which leads to secretion of α-granules and exposure of the fibrinogen-binding receptor, integrin α_IIb_β_3_ (GPIIb/GPIIIa), on the surface of platelets. VaaMPIII-3 might inhibit activation of ADP-stimulated platelets as an antagonist of their ADP receptors. We excluded the possibility that VaaMPIII-3 affects the later stages of the ADP-induced aggregation process by showing that it does not interact with GPIIb or GPIIIa (α_IIb_β_3_) [49,50].

The DC proteins of different origins are potent inhibitors of collagen-induced platelet aggregation [12,13,16,17,26,27,47,48,51,52]. VaaMPIII-3 had the same effect. Its potency of 80% inhibition of collagen-induced platelet aggregation at 700 nM is similar to that of halysetin from the venom of *Agkistrodon halys* (IC_50_, 420 nM) [13] and A/C (IC_50_, 456 nM) [48], but lower than that of catrocollastatin-C (IC_50_, 66 nM) [17], jararhagin-C (IC_100_, 200 nM) [12], and A/DC (IC_50_, 110 nM) [47]. Again, the difference between the effects of a DC protein and its C domain alone (e.g., A/DC and A/C) suggests that the D domain also has an important role in inhibition of collagen-induced platelet aggregation [47,48]. The most widely recognized structural determinant of this kind of inhibitory action in the D domain of DC proteins is the SECD motif, which is present in VaaMPIII-3 [28,31,53]. However, the structural determinants of inhibition of collagen-induced platelet aggregation in the C-domain of DC proteins have not been defined, although they are probably located in the HVR [31,54]. Therefore, we propose that VaaMPIII-3 inhibits collagen-induced platelet aggregation mainly through the interaction of its SECD motif with integrin α_2_β_1_ on the surface of platelets, although possibly also through the interaction of its HVR with different platelet receptors [31]. Using flow cytometry, we ruled out the possibility of its action being mediated by binding to GPVI, another platelet receptor that is important in collagen-induced aggregation. As the D domain of VaaMPIII-3 is truncated, it is possible that there is another yet-unknown functional region in the D domain of other DC proteins that supports the interaction with platelet receptors that is absent in VaaMPIII-3, thus resulting in its reduced inhibitory effect. Given the sequence similarity of VaaMPIII-3 with other DC proteins that have been shown to bind collagen (e.g., A/C, catrocollastatin-C, HF3-DC), another possible mechanism for this inhibitory effect is that VaaMPIII-3 binds to collagen and prevents its interactions with integrin receptors on platelets [17,43,52].

Inhibition of arachidonic-acid-induced platelet aggregation has been tested for only a few DC proteins. Arachidonic acid activates platelet aggregation indirectly by first being converted to thromboxane A2, which then activates the neighbouring platelets by binding to thromboxane-prostanoid receptors [49]. VaaMPIII-3 reduced arachidonic-acid-induced platelet aggregation by 70% at 700 nM. Thus, its inhibitory effect is less potent than that of leberagin-C (IC_50_, 50 nM) [11]. The mechanism of inhibition of arachidonic-acid-induced aggregation by leberagin-C and VaaMPIII-3 has not been studied, but we speculate that they can act as antagonists of thromboxane-prostanoid receptors.

Some DC proteins isolated from *Bothrops* snake venoms, such as BmooPAi and BaltDC, can interfere with ristocetin-induced vWF-dependent platelet agglutination [14,18]. They have been proposed to act as antagonists of the GPIb platelet receptor of vWF. They act specifically because they do not inhibit ADP- or collagen-induced platelet aggregation. However, VaaMPIII-3 did not affect vWF-dependent platelet agglutination, and it did not interfere with the GPIX subunit of the vWF-binding receptor complex. On the other hand, a decrease in MFI was observed for CD42b, which suggests VaaMPIII-3 binding to the GPIb-α receptor. Considering the results of ristocetin-induced vWF-dependent platelet agglutination, which were determined at considerably higher platelet concentrations, it appears that binding to the GPIb-α receptor has only a limited impact in the pathogenesis of VaaMPIII-3 in vivo. The absence of this activity in our sample also confirms the absence of possible contamination with residual disintegrins, which are potent inhibitors of platelet agglutination [55].

## 4. Conclusions

We have shown that the pre-pro-protein precursor of VaaMPIII-3 is transcribed and translated from its own gene. This gene evolved from the ancestral P-III class SVMP gene after duplication and subsequent deletion of the MP domain and the first part of the D domain coding gene region. In our opinion, the characteristics of evolution of VaaMPIII-3 and related proteins justify the classification of these proteins into a new P-IIIe subclass of SVMPs, even though they lack the metalloproteinase domain.

We isolated the first P-IIIe SVMP-derived representative protein, VaaMPIII-3, and showed that it has similar biochemical characteristics as other DC proteins. It inhibits platelet aggregation induced by ADP, collagen, and arachidonic acid with similar efficacies.

We constructed the 3D homology model of VaaMPIII-3, and based on this, we proposed the possibility of disulphide bond formation between either the Cys6 residue and the Cys26 residue, or the Cys19 residue and the Cys26 residue. Alternative disulphide bonds might lead to significant conformational differences for the protein, especially in the part where the integrin-binding motif SECD is positioned. This might alter the activity of the protein, so we propose that an intramolecular thiol-disulphide exchange or a disulphide switching mechanism is involved in the regulation of at least part of the biological activities of VaaMPIII-3.

Thrombotic diseases continue to be a major health problem in the modern world, and natural molecules are being intensively sought to develop more efficient and specific antithrombotic drugs. VaaMPIII-3 is an interesting molecule in this regard, especially because of the possibility of controlling its action via the redox potential.

## 5. Materials and Methods

### 5.1. Materials

Genomic DNA was isolated from a *Vaa* specimen that originated from north-western Slovenia, near the border with Austria. Crude *Vaa* venom was obtained from the Institute of Immunology, Inc., Zagreb, Croatia. All of the other chemicals were of analytical, sequencing, or mass spectrometry grade.

### 5.2. Isolation of VaaMPIII-3 from Crude Venom of the Nose-Horned Viper

VaaMPIII-3 was purified from the crude lyophilized *Vaa* venom by gel filtration, cation-exchange, and covalent chromatography. Crude venom was dissolved in 20 mM Tris buffer, pH 7.0, containing 300 mM NaCl and 2 mM CaCl_2_, and applied to a Superdex 75 10/300 GL column (24 mL) equilibrated in the same buffer. Fraction B2 from the gel filtration was dialysed against 20 mM MES buffer with 2 mM CaCl_2_, pH 6, and applied to an equilibrated SP Sepharose Fast Flow cation-exchange column (35 × 1.6 cm). The unbound fraction was dialysed against 100 mM Tris buffer, pH 7.4, containing 300 mM NaCl and 1 mM EDTA, activated with 5 mM DTT, applied to an activated Thiol-Sepharose 4B column (5 mL), and incubated overnight at 4 °C, with shaking. The bound fraction was eluted with 20 mM DTT, concentrated using a 10-kDa-cut-off membrane, and then dialysed against assay buffer (20 mM HEPES, pH 7.4, 50 mM NaCl, 2 mM CaCl_2_).

### 5.3. SDS-Polyacrylamide Gel Electrophoresis

SDS-polyacrylamide gel electrophoresis (SDS-PAGE) of protein samples on 12.5% (*w*/*v*) polyacrylamide gels was performed under reducing and non-reducing conditions, as previously described [56]. Molecular mass standards were from Fermentas (Thermo Fisher Scientific, Vilnius, Lithuania).

### 5.4. Isoelectric Focusing

Isoelectric focusing was performed on a Phast System (Amersham Pharmacia Biotech, Uppsala, Sweden) using PhastGel 3-9 slab-gel (0.35 × 43 × 50 mm) and pI standards (3.5–9.3) from the same manufacturer. Proteins were visualised using Coomassie Brilliant Blue R250 or silver staining.

### 5.5. Two-Dimensional Gel Electrophoresis

Five µg of VaaMPIII-3 was analysed by two-dimensional gel electrophoresis as previously described [10], using a rehydration buffer without detergents and running the second dimension SDS-PAGE at 10 mA per gel.

### 5.6. Reversed Phase High-Performance Liquid Chromatography

VaaMPIII-3 was analysed on a C18 RP-HPLC column (BIOshell A400 Protein C18 Column 5 cm × 75 μm, 3.4 μm particle size; BIOshell Teoranta, Carrowteige, Ballina, Ireland) using a 5-min linear gradient from 0% to 30% and a 10-min linear gradient from 30% to 40% (*v*/*v*) of acetonitrile in 0.1% *trifluoroacetic acid* (*v*/*v*) at 1 mL/min. Proteins were detected at 215 nm, collected manually, lyophilised, and stored at −20 °C until SDS-PAGE and sequence analysis.

### 5.7. N-Terminal Amino Acid Sequencing

N-terminal amino acid sequencing of purified VaaMPIII-3 was performed by automated Edman degradation on a Procise 492A Automated Sequencing System (Applied Biosystems, Waltham, MA, USA) and a PPSQ-53A Gradient System (Shimadzu, Kyoto, Japan).

### 5.8. Mass Spectrometry

VaaMPIII-3 (5 µg) was first separated by SDS-PAGE on 12% (*w*/*v*) polyacrylamide gels. The protein bands were cut out, trypsinised, and subjected to mass spectrometry analysis using a 1200 series HPLC-Chip-LC/MSD Trap XCT Ultra ion trap mass spectrometer (Agilent Technologies, Waldbronn, Germany), as previously described [57]. MS/MS spectra were analysed as previously described [10].

### 5.9. Protein N-deglycosylation

Four µg of vacuum-dried VaaMPIII-3 were denatured and treated with 3 U peptide *N*-glycosidase F (Roche, Mannheim, Germany), as previously described [55].

### 5.10. Determination of the Position of the Free Cys Residue

VaaMPIII-3 (10 µg) was carboxymethylated with iodoacetamide (9.5 mM) under mild reducing conditions (1.5 mM DTT). The free Cys residue was identified as CM-Cys by Edman sequencing.

### 5.11. Determination of the Thermal Stability of the Protein

The thermal stability of VaaMPIII-3 was assessed under different conditions using DSF, as previously described [55]. The final volume of the experimental mixture was 10 μL, and it was composed of 6 μL of the relevant buffer, 2 μL protein solution (2 μM final concentration), and 2 μL 40-fold water-diluted SYPRO Orange stock solution (Sigma, Schnelldorf, Germany). The change in fluorescence of SYPRO Orange was recorded in the temperature range of 25 °C to 90 °C, increasing at a rate of 0.5 °C/min. The protein melting temperature (Tm) was calculated from the first derivative of the measured fluorescence signals using the online research tool DSFWorld [58].

### 5.12. Blood Coagulation Assays

The activated partial thromboplastin time (aPTT) and prothrombin time (PT) were measured as previously described [3]. We used human pooled plasma (Pool Norm, Diagnostica Stago, Asnieres, France) and HemosIL reagents—SynthASil and HemosIL CaCl_2_ 0.020 M (Instrumentation Laboratory, Bedford, MA, USA)—for aPTT, and ReadiPlasTin (Instrumentation Laboratory, Bedford, MA, USA) for the PT assay. The effect of 1 µM VaaMPIII-3 was assessed on a BCT system (Dade Boehring, Marburg, Germany) according to the respective assay protocol. Results are expressed as proportions (in percentages) as means ± SEM of duplicate measurements of the relative deviation from the control value.

### 5.13. Platelet Aggregation and Agglutination Assays

The effects of VaaMPIII-3 on platelet aggregation induced by ADP, collagen and arachidonic acid, and platelet agglutination induced by ristocetin, were studied as described previously [55]. Human blood was donated by the experimenter according to permission no. 53/08/11 of the National Medical Ethics Committee of the Republic of Slovenia. Platelet-rich plasma was obtained by 10-min centrifugation of citrated blood at 1500× *g* without applying the brake. The platelet count of the carefully removed platelet-rich plasma was 200 × 10^9^/L. VaaMPIII-3 was tested at 700 nM, 350 nM, 180 nM and 70 nM.

### 5.14. Platelet Receptor Binding Assays

Binding of VaaMPIII-3 to various platelet receptors (i.e., GPIb-α, GPIIb, GPIIIa, GPIX, GPVI) was tested by flow cytometer (Navios; Beckman Coulter, Brea, CA, USA) using a 488-nm laser. Platelet-rich plasma was obtained as described in Section 5.13, and the platelet count was adjusted to 15 × 10^9^/L by addition of phosphate-buffered saline. Here, 2 µL (1 µM final concentration) VaaMPIII-3 or buffer (control) was added to 20 µL platelet suspension and incubated for 10 min. Then, 5 µL platelet receptor antibody was added, with incubation for 25 min at room temperature in the dark. FITC-conjugated antibody CD41 (GPIIb or α_IIb_), CD42a (GPIX), CD42b (GPIb-α) (Immunotech, Beckman Coulter, Marseille, France), or PE-conjugated antibody CD61 (GPIIIa or β_3_) (Immunotech, Beckman Coulter, Marseille, France), and GPVI (Becton Dickinson, Franklin Lakes, NJ, USA) were used. Before measurement, 500 µL phosphate-buffered saline was added to each sample. The MFI value of the control sample was compared to the MFI values of the samples containing VaaMPIII-3. The effects of VaaMPIII-3 are presented as proportions (as percentages) of available antigen in comparison with the measurement in which a non-platelet receptor antibody was used (MsIgG isotype control; i.e., FITC or PE conjugated MsIgG).

### 5.15. Homology Modelling

A model of the 3D structure of VaaMPIII-3 was generated using the Modeller 10v1 homology modelling software [59]. Basic Local Alignment Search Tool (BLAST) [60] was used to identify the sequence most similar to VaaMPIII-3, the experimentally determined 3D structure of which is deposited in the Protein Data Bank (PDB). The 3D structure of AaHIV was extracted (PDB code: 3HDB) as a metalloproteinase from the venom of *A. acutus*. This AaHIV was used as the modelling template, which is 67% identical and 83% similar in the corresponding part of the amino acid sequence to the sequence of VaaMPIII-3. The sequences of these proteins were aligned using ClustalO [61] and adjusted manually. Modelling was performed using default parameters, the ‘allHmodel’ protocol to include hydrogen atoms, and the ‘HETATM’ protocol to include Ca^2+^ ions. The model with the lowest molpdf value (probability density function) and the best stereochemical quality (validated with PROCHECK [62]) was selected.

## Figures and Tables

**Figure 1 toxins-14-00232-f001:**
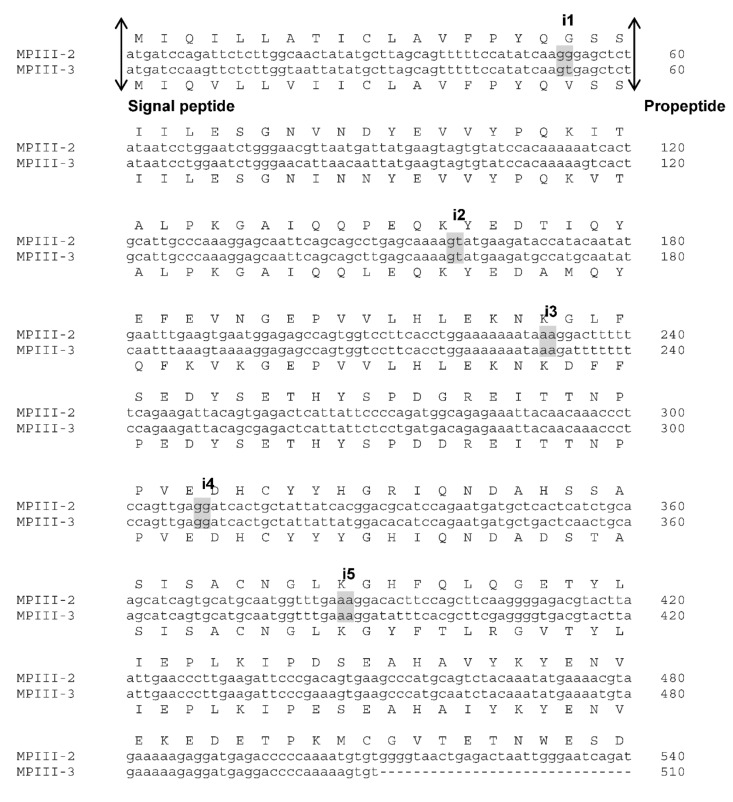
Alignment of the nucleotide and deduced amino acid sequence of pre-pro-VaaMPIII-2 with that of VaaMPIII-3. The beginning of the signal peptide, the propeptide, the MP (metalloproteinase) domain, the D (disintegrin-like) domain, the mature VaaMPIII-3 protein, and the C (cystein-rich) domain are indicated by the up and down arrows. Deletions are indicated by dashes, and intron positions in the VaaMPIII-2 gene are indicated above the two shaded adjacent nucleotides. Introns 1 to 6 in the VaaMPIII-2 gene correspond to introns 1 to 6 in the VaaMPIII-3 gene, introns 7 to 13 are absent in the VaaMPIII-3 gene, and introns 14 to 16 in the VaaMPIII-2 gene correspond to introns 7 to 9 in the VaaMPIII-3 gene. C^+^, Cys residue is believed to be involved in the S-S-linked two-chain snaclec subunit in VaaMPIII-2, the P-IIId subclass SVMP. D-loop, disintegrin-like loop or XXCD motif, i.e., an RGD-like motif; i, intron; *, stop codon.

**Figure 2 toxins-14-00232-f002:**
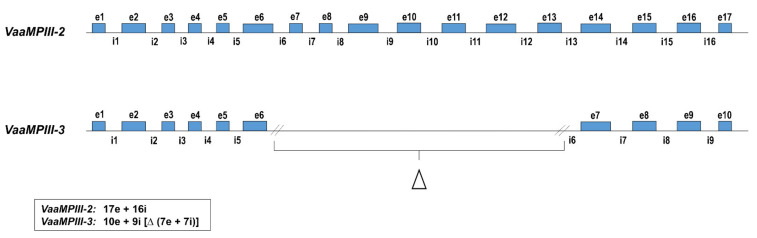
Schematic diagram of the homologous VaaMPIII-2 and VaaMPIII-3 gene structures. Exons are indicated by blue boxes and introns by lines. Lengths of introns are not shown according to their actual nucleotide sequence lengths. Δ, deletion; e, exon; i, intron.

**Figure 3 toxins-14-00232-f003:**
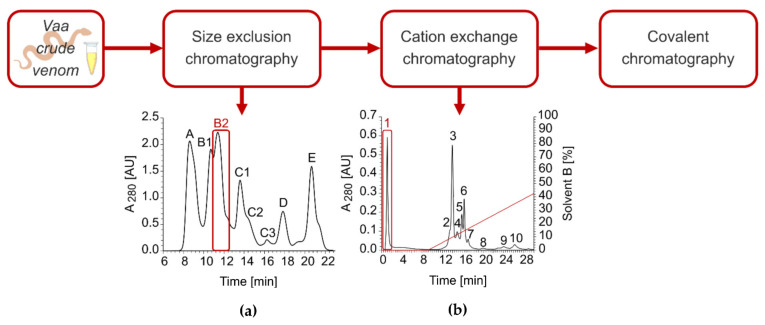
Procedure to purify VaaMPIII-3. Crude *Vaa* venom (1 g) was dissolved in 20 mM Tris buffer, pH 7.0, containing 300 mM NaCl and 2 mM CaCl_2_, and applied to a Superdex 75 10/300 GL size-exclusion chromatography column equilibrated with the same buffer (**a**). At a flow rate of 0.5 mL/min, fractions of 1 mL were collected. Fraction B2 contained proteins of 20–30 kDa, which were further separated by cation-exchange chromatography (**b**). The B2 fraction was dialysed against 20 mM MES buffer, pH 6, containing 2 mM CaCl_2_, and then applied to an SP Sepharose Fast Flow column equilibrated with the same buffer. The analysis was operated at 1 mL/min, with 1 mL fractions collected. The red line represents the gradient of buffer B (0–2 M NaCl in 20 mM MES, pH 6, with 2 mM CaCl_2_). VaaMPIII-3 was eluted in unbound fraction 1. Pure VaaMPIII-3 was obtained after the third step of covalent chromatography on Thiol-Sepharose 4B, taking advantage of the unpaired Cys residue in VaaMPIII-3.

**Figure 4 toxins-14-00232-f004:**
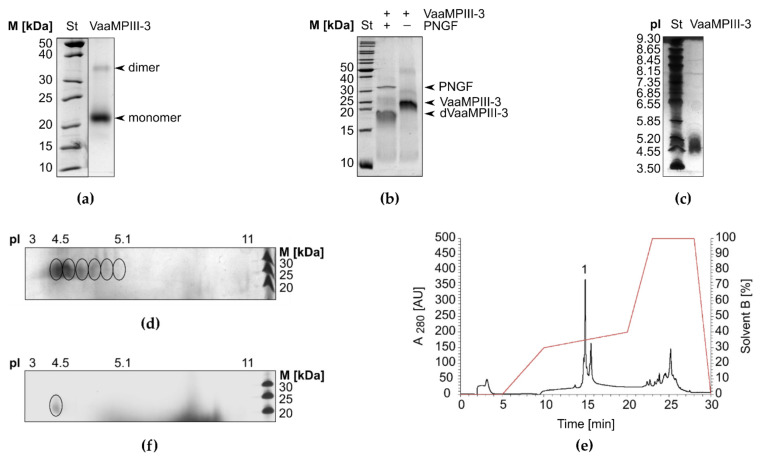
Physicochemical characterisation of VaaMPIII-3. (**a**) SDS-PAGE analysis under non-reducing conditions showed that VaaMPIII-3 is a 21-kDa glycoprotein that can form a dimer. (**b**) *N*-deglycosylation of VaaMPIII-3 by peptide *N*-glycosidase F (PNGF) resulted in a ~17-kDa product (dVaaMPIII-3). (**c**) Isoelectric focusing showed that VaaMPIII-3 has a heterogeneous pI. (**d**) Two-dimensional gel electrophoresis revealed that the VaaMPIII-3 sample consisted of six isoforms (encircled). (**e**) The predominant VaaMPIII-3 isoform was separated from the others on a C18-RP-HPLC column equilibrated in 0.1% (*v*/*v*) *trifluoroacetic acid* and eluted at a flow rate of 0.8 mL/min with a gradient of solvent B (0–90% (*v*/*v*) acetonitrile in 0.1% (*v*/*v*) *trifluoroacetic acid*) as indicated (red line). (**f**) Two-dimensional gel electrophoresis of RP-HPLC peak 1 demonstrated the presence of just one isoform of VaaMPIII-3, the most abundant one, with pI of 4.5.

**Figure 5 toxins-14-00232-f005:**
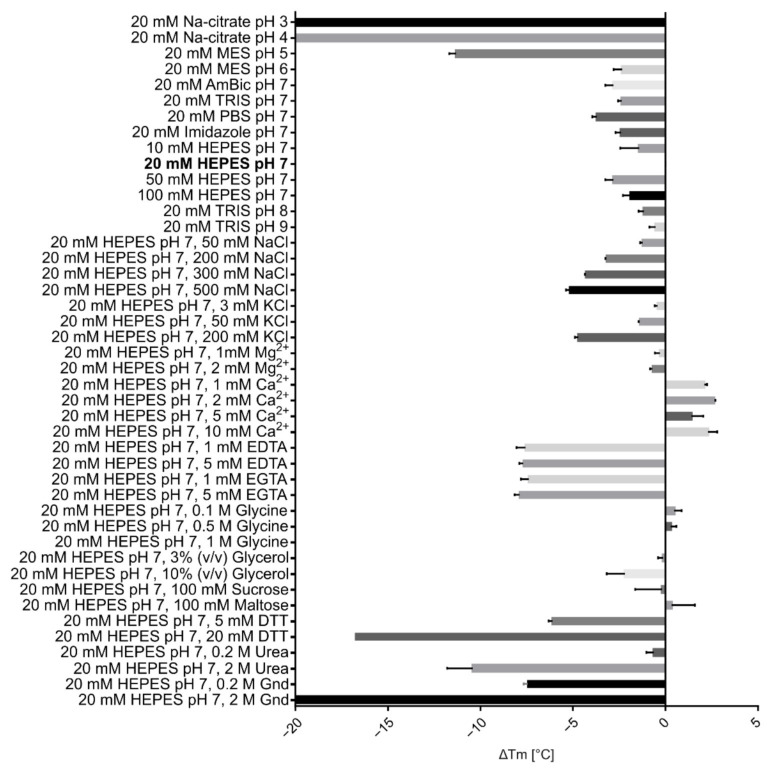
Stability of VaaMPIII-3 in buffers of different compositions. Differential scanning fluorimetry (DSF) was used to determine the compositions of the buffers that optimally preserved the native structure of VaaMPIII-3. Using DSF, the thermostability of a protein in a given environment is characterised by the melting temperature (Tm), the temperature at which it unfolds or melts. The reference condition was 20 mM HEPES buffer at pH 7 (bold). The effects of buffer exchange or of an additive to the buffer solution on the thermostability of VaaMPIII-3 is reflected in the shift of the protein Tm from the Tm measured under the reference condition (ΔTm). A negative ΔTm indicates that a particular additive decreased the thermostability of VaaMPIII-3 in comparison to its stability under the reference condition, and vice versa. AmBic, ammonium bicarbonate; DTT, dithiothreitol; Gnd, guanidine hydrochloride.

**Figure 6 toxins-14-00232-f006:**
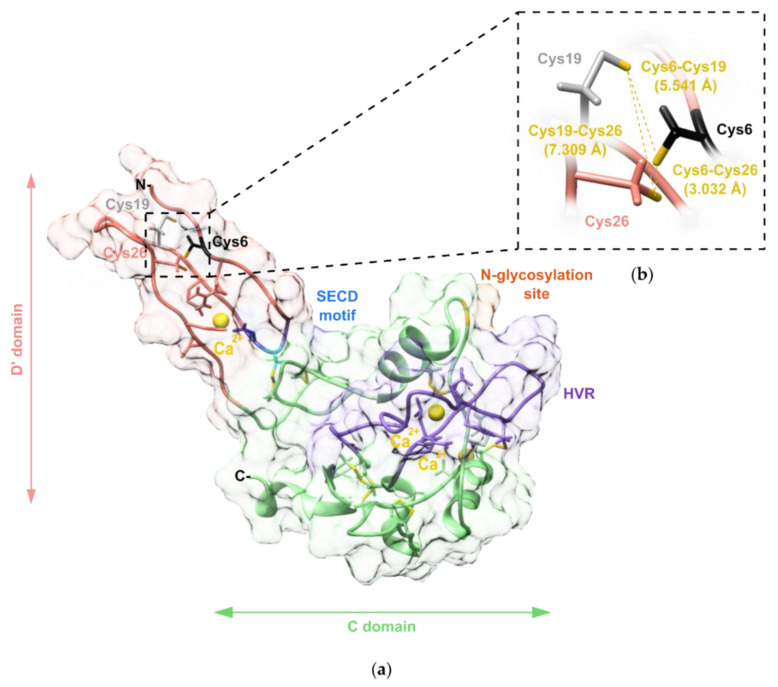
The three-dimensional homology model of VaaMPIII-3. (**a**) The structural model of VaaMPIII-3 prepared by homology modelling based on the crystal structure of AaHIV (PDB code: 3HDB). The truncated disintegrin-like (D’) domain is dark orange and the cysteine-rich (C) domain is green. Cys6, the integrin-binding motif, SECD, the *N*-glycosylation site, and the hypervariable region (HVR) are shown in black, blue, orange, and purple, respectively. Ca^2+^ ions are shown as yellow spheres, and disulphide bonds as yellow sticks. Cys6, Cys19 and Cys26 are shown in the thiol form due to the possibility of alternative pairing between these Cys residues. In AaHIV, the Cys267-Cys287 bond (corresponding to the Cys6-Cys26 bond in VaaMPIII-3) is formed, and Cys280 (Cys19 in VaaMPIII-3) pairs with Cys254 in the N-terminal part of the D domain, which is absent in VaaMPIII-3. (**b**) Distances between Cys6, Cys19 and Cys26 residues in the D’ domain of VaaMPIII-3 calculated from the model, to allow for formation of either a Cys6-Cys26 or a Cys19-Cys26 bond. The existence of both unpaired Cys6 and Cys19 was experimentally confirmed by N-terminal sequencing of the native VaaMPIII-3, which supports the hypothesis of Cys switching.

**Figure 7 toxins-14-00232-f007:**
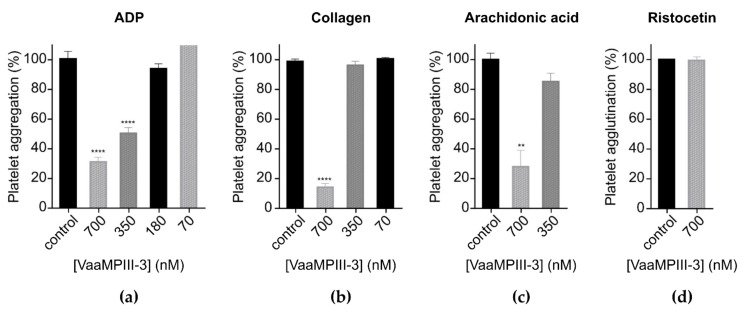
Effects of VaaMPIII-3 on platelets. Platelet aggregation induced by ADP (**a**), collagen (**b**), and arachidonic acid (**c**) was significantly inhibited by VaaMPIII-3, whereas von Willebrand factor (vWF)-dependent platelet agglutination induced by ristocetin (**d**) was not. Control values were determined in the absence of VaaMPIII-3 and interpreted as 100% change in the optical density of the assay solution. The data are expressed relative to the control values and are means ± SEM of at least three measurements. ** *p* < 0.01, **** *p* < 0.0001 (one-way ANOVA followed by Tukey’s tests, using GraphPad Prism).

**Table 1 toxins-14-00232-t001:** The contents of secondary structure elements in VaaMPIII-3. To assess the quality of the 3D homology model of VaaMPIII-3, the contents of the different types of secondary structure elements were calculated from the CD spectrum predicted from its 3D model using the PDBMD2CD web server and compared with the experimentally obtained values.

Secondary Structure Element	Secondary Structure Contents (%)	
	3D Model	Experimental Data
α-Helix	14.8	10.9
Antiparallel β-sheet	27.8	27.4
Turn	13.1	15.8
Other	44.2	45.9

## Data Availability

Data is contained within the article or Appendix A.

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
