# Peer review of "Genomic Confirmation of the P-IIIe Subclass of Snake Venom Metalloproteinases and Characterisation of Its First Member, a Disintegrin-Like/Cysteine-Rich Protein"

_toxins, 2022, doi:10.3390/toxins14040232_

Round 1

Reviewer 1 Report

The authors have shown from SVMP genome of Vipera ammodytes ammodytes that the disintegrin-like / cysteine-rich (DC) protein, a preproprotein precursor of VaaMPIII-3, is transcribed and translated from its own gene. They also showed that this gene was predicted to evolve from the ancestral P-III class SVMP gene after replication and subsequent deletion of the first part of the MP and D domain coding gene regions. Analysis of the 3D model also suggested that intramolecular thiol-disulfide exchange may have a regulatory function. It is very interesting because it clarifies the unclear points of the research on DC so far.

Reviewers believe that the following modifications will help readers understanding.

Major points

In the discussion section, please describe the limits of the content considered from the experimental results obtained in this paper as a separate paragraph.

Minor points

The following description in the Results in the paper seems to be incorrect, so please correct it.

2.4. Next to Modeling of the Three-Dimensional Structure of VaaMPIII-3

2.4. Determination of the Position of the Unpaired Cys Residue in VaaMPIII-3, using the same number as the subtitle.

Reviewer 2 Report

The manuscript “Genomic Confirmation of the P-IIIe Subclass of Snake Venom Metalloproteinases and Characterisation of Its First Member, a Disintegrin-Like/Cysteine-Rich Protein” describes the purification and characterization of VaaMPIII-3, a new snake venom metalloprotease (SVMP) subclass P-IIIe desintegrin-like/ cysteine-rich (DC) protein from European nose-horned viper Vipera ammodytes ammodytes snake venom. Authors purified the protein from the crude venom using three different chromatography techniques: gel filtration, cation-exchange chromatography, and covalent binding to Thiol-Sepharose 4B; Edman sequencing and ESI-MS/MS sequencing analysis. Furthermore, authors characterized the protein using SDS-PAGE, 2D- gel electrophoresis, RP-HPLC, structural thermostability analysis using differential scanning fluorimetry, molecular 3D structure modelling, including S-S interaction determination between specific Cys-residues, and functional characterization of platelet aggregation inhibition induced by ADP, collagen and arachidonic acid.

I believe that the authors were able to successfully describe the characterization of the VaaMPIII-3, including the purification steps, and biochemical, structural and functional characterization of the protein. Discussion is well developed and authors were able to give a good overview of this class of protein and contextualize it based on the literature.

Just as a minor point, although the manuscript is well structured and the English is good, it would benefit from a revision from an English Native Speaker.

Reviewer 3 Report

The manuscript entitled "Genomic Confirmation of the P-IIIe Subclass of Snake Venom Metalloproteinases and Characterisation of Its First Member, a Disintegrin-Like/Cysteine-Rich Protein" describes the chromatographic purification and analysis of the Snake Venom Metalloproteinases (DC protein) VaaMPIII-3 from Vipera ammodytes ammodytes (Vaa). Also, this work shows the genomic analysis to demonstrate the origin of DC protein, taking VaaMPIII-3 as an example. The authors found that VaaMPIII-3 is transcribed and translated from its own gene, rather than being originated by proteolytic digestion of a larger SVMP. Based on these results, the authors propose introducing a new subclass of SVMPs that groups VaaMPIII-3 and related proteins.

The manuscript is well-written, within the journal's scope, and should be interesting to the general reader of Toxins.

However, I find some lack of information regarding the analysis of the isolated VaaMPIII-3. I suggest to add some (supplementary) figures related to the identification of VaaMPIII-3:

1- The authors used SDS-PAGE and in-gel digestion to identify the molecule. A figure that illustrates the fragments identified and sequence coverage would be helpful.

2- SDS-PAGE provides some idea of the molecular mass of a protein, but this is very far from the accuracy provided by mass spectrometry. However, no mass spectrum is shown to illustrate that the protein mass matches the one expected from the sequence. I understand that glycosylation can make this problematic, however the authors used peptide N-glycosidase F to removed the glycosylation, so in principle, the protein without glycosylation could be measured by mass spectrometry to confirm the presence of the full-length protein. I suggest to include an HR mass spectrum of the protein after glycans removal and an explanation accordingly.

3-The authors suggest that Cys9 or Cys19 may form a disulfide bridge with Cys26, according to the 3D model. I think an HR mass spectrum (the same as in point 2) of the molecule without glycans would be very helpful to confirm the existence of that disulfide bridge.  

Author Response

Dear Reviewer,

We are glad that you expressed a very positive opinion about our manuscript (toxins-1629264). You suggested some modifications and we thank you for your constructive remarks. Below are our responses to your queries and the description of our actions, point-by-point. For easier following of our modifications in the revised version of the manuscript the WORD TRACK CHANGES functionality has been used.

You found some lack of information regarding the analysis of the isolated VaaMPIII-3 and suggested addition of some (supplementary) figures related to the identification of VaaMPIII-3:

> 1- The authors used SDS-PAGE and in-gel digestion to identify the molecule. A figure that illustrates the fragments identified and sequence coverage would be helpful.

Our response: To supplement the description of the isolated VaaMPIII-3, we created the Supplementary Figure S2 that illustrates the fragments identified and displays the sequence coverage. The new Figure is mentioned in line 292 of our manuscript.

> 2- SDS-PAGE provides some idea of the molecular mass of a protein, but this is very far from the accuracy provided by mass spectrometry. However, no mass spectrum is shown to illustrate that the protein mass matches the one expected from the sequence. I understand that glycosylation can make this problematic, however the authors used peptide N-glycosidase F to removed the glycosylation, so in principle, the protein without glycosylation could be measured by mass spectrometry to confirm the presence of the full-length protein. I suggest to include an HR mass spectrum of the protein after glycans removal and an explanation accordingly.

Our response: To supplement the description of the isolated VaaMPIII-3, we added its mass specter as the Supplementary Figure S3. The additional result is mentioned in line 309 of our manuscript.

> 3-The authors suggest that Cys9 or Cys19 may form a disulfide bridge with Cys26, according to the 3D model. I think an HR mass spectrum (the same as in point 2) of the molecule without glycans would be very helpful to confirm the existence of that disulfide bridge.

Our response: As the molecular masses of VaaMPIII-3 containing either Cys9-Cys26 or Cys19-Cys26 are identical, the HR mass spectrometry cannot discriminate between the two forms

Round 2

Reviewer 3 Report

1-The authors have successfully tackled the first comment by showing the fragments sequenced by Edman or MS/MS.

2-Regarding the second comment: I should say that the authors showed a mass spectrum of the glycosylated molecule. It looks nice but I had actually suggested to provide an HR mass spectrum of the molecule without glycosylation, aiming to match the experimental mass with the theoretical one calculated from the sequence. That mass spectrum would be helpful either for point 2 or 3.

3-Regarding the third comment: Of course, mass spectrometry cannot distinguish between Cys9-Cys26 or Cys19-Cys26 solely from the monoisotopic mass of the molecule. However, my request was about showing an HR mass spectrum (molecule without glycosylation) that confirmed the formation of that disulfide bridge, regardless of how it was formed, as a support for the bioinformatic prediction. 

Nonetheless, in defense of this nice work, I must say that the figure showing the sequence coverage is helpful to cover point 2, considering the purpose of the work. Regarding point 3, the formation of that disulfide bridge is just a prediction, and the authors do not go beyond, so it is not mandatory to demonstrate it. I think this is acceptable.  

Minor comment:
Figure S2. Is the NCBI accession number correct? Please check it.
MG958498 is from UniProtKB -A0A6B7FRK6 (A0A6B7FRK6_VIPAA). The sequence in this entry is different from the one in the figure. 
The sequence shown in the figure is located here: UniProtKB - A0A6B7FMR5 (A0A6B7FMR5_VIPAA)

Author Response

Please see the attached manuscript and supplementary file